# Minimax Risks and Optimal Procedures for Estimation under Functional Local Differential Privacy

**Bonwoo Lee**         **Jeongyoun Ahn**[*]         **Cheolwoo Park**[*]
Korea Advanced Institute of Science & Technology
Daejeon, 34141 South Korea
righthim@kaist.ac.kr, jyahn@kaist.ac.kr, parkcw2021@kaist.ac.kr

## Abstract

As concerns about data privacy continue to grow, differential privacy (DP) has emerged as a fundamental concept that aims to guarantee privacy by ensuring individuals' indistinguishability in data analysis. Local differential privacy (LDP) is a rigorous type of DP that requires individual data to be privatized before being sent to the collector, thus removing the need for a trusted third party to collect data. Among the numerous (L)DP-based approaches, functional DP has gained considerable attention in the DP community because it connects DP to statistical decision-making by formulating it as a hypothesis-testing problem and also exhibits Gaussian-related properties. However, the utility of privatized data is generally lower than that of non-private data, prompting research into optimal mechanisms that maximize the statistical utility for given privacy constraints. In this study, we investigate how functional LDP preserves the statistical utility by analyzing minimax risks of univariate mean estimation as well as nonparametric density estimation. We leverage the contraction property of functional LDP mechanisms and classical information-theoretical bounds to derive private minimax lower bounds. Our theoretical study reveals that it is possible to establish an interpretable, continuous balance between the statistical utility and privacy level, which has not been achieved under the $\epsilon$-LDP framework. Furthermore, we suggest minimax optimal mechanisms based on Gaussian LDP (a type of functional LDP) that achieve the minimax upper bounds and show via a numerical study that they are superior to the counterparts derived under $\epsilon$-LDP. The theoretical and empirical findings of this work suggest that Gaussian LDP should be considered a reliable standard for LDP.

## 1 Introduction

It has been widely accepted that anonymization is insufficient in protecting privacy (Sweeney, 2000, 2002; Dinur and Nissim, 2003). The concerns about data privacy have grown significantly, particularly with the advancement of computer science technology and the rise in data generated by individuals and tech companies. Such concerns are shared by politics and industry, leading to the adoption of France's "Loi pour une République numérique (Law for the Digital Republic)" in October 2016 (Algan et al., 2016), EU's General Data Protection Regulation in May 2018, and the California Consumer Privacy Act (Wang et al., 2022), all of which regulate data protection, collection, and processing. Also, data privacy techniques have been implemented in industries such as Google (Erlingsson et al., 2014; Fanti et al., 2016), Apple (Thakurta et al., 2017; Tang et al., 2017), Microsoft (Ding et al., 2017), and SAP (Kessler et al., 2019).

---

[*]Corresponding authors

37th Conference on Neural Information Processing Systems (NeurIPS 2023).

Differential privacy (DP), suggested by Dwork et al. (2006b), has become a fundamental foundation of the modern privacy concept. Its core idea is that privacy can be protected not by how well the sensitive information is hidden but rather by how individuals are indistinguishable in the privatized data. Dwork et al. (2006b) first introduced the notion of $\epsilon$-DP for a randomized mechanism that yields a likelihood ratio between its outputs from 'neighboring' data that is bounded by a privacy budget $\epsilon$. Despite its intuitive appeal, $\epsilon$-DP has been found to lack generality in explaining some common privacy mechanisms including the Gaussian mechanism (Mironov, 2017; Dong et al., 2022). A generalized version of $\epsilon$-DP, $(\epsilon, \delta)$-DP, is introduced in Definition 1.1. Let $\mathcal{X}$ be the sample space and $M : \mathcal{X}^n \longrightarrow \mathcal{P}(\mathcal{Z})$ be a randomized mechanism, where $\mathcal{P}(\mathcal{Z})$ is a family of distributions over $\mathcal{Z}$. Let $S \sim S'$ denote that data sets $S$ and $S'$ are neighbors, differing only by one individual. The following definition states that the output distribution of $M$ does not heavily depend on the presence of a specific individual. Note that $(\epsilon, 0)$-DP reduces to $\epsilon$-DP.

**Definition 1.1** ($(\epsilon, \delta)$-DP, Dwork et al. (2006a)). A mechanism $M$ is $(\epsilon, \delta)$-DP if the following holds:

$$\sup_{S,S' \subset \mathcal{X}, S \sim S'} \sup_{A \subset \mathcal{Z}} \mathbb{P}\left(M(S) \in A\right) - e^\epsilon \mathbb{P}\left(M(S') \in A\right) \leq \delta.$$

Another type of DP, local differential privacy (Duchi et al., 2013, LDP), provides strong privacy guarantees by privatizing raw data before releasing it to a data collector. It is commonly used for private statistical inference since it gives the adversary access to privatized data that is the same size as the original data. Note that unlike in Definition 1.1, the inputs of LDP mechanisms are not the whole data set $S$ but each individual observation $x_i \in S$.

As expected, the accuracy of statistical analysis performed on privatized data is often worse than that of non-private data. This has prompted researchers to seek a balance between privacy and utility as well as to find the optimal mechanism given privacy constraints. The efforts to balance noise contamination and utility date back to Carroll and Hall (1988), which predated the emergence of DP. Later, Duchi et al. (2013) proposed a framework for quantifying the trade-off between privacy and statistical utility using $\epsilon$-LDP and minimax risk analysis. With the minimax framework, one can study the minimum loss that can be attained in the worst-case scenario for statistical utility, given a specific privacy constraint. Since then, numerous researchers have explored minimax risks for various inference problems under $\epsilon$-LDP (Li et al., 2022; Chhor and Sentenac, 2023; Rohde and Steinberger, 2020; Butucea et al., 2020).

However, despite the inherent continuity in the definitions of $\epsilon$-LDP and $(\epsilon, \delta)$-LDP, some studies on LDP mechanisms have revealed an inexplicable discrepancy in the statistical utility achieved by the two techniques (Asoodeh et al., 2021). One possible explanation for this phenomenon is the inefficiency of the composition rule used in $(\epsilon, \delta)$-LDP, which has been found to perform poorly in tracking privacy leakage resulting from the compositions of multiple mechanisms or subsampling, as pointed out by some studies including Dong et al. (2022); Mironov (2017). Since data are frequently reused multiple times in most data analysis scenarios, the privacy level must be calculated by taking the composition of multiple mechanisms into account. An accurate assessment of privacy under such compositions is critical, since underestimating privacy would force a mechanism to sacrifice utility to ensure the desired privacy level.

This work explores the trade-off between statistical utility and local privacy under the framework of functional differential privacy (Dong et al., 2022, FDP), which is known to provide more precise privacy control. The core idea of FDP connects the fundamental DP concept, which involves making two outputs based on different inputs indistinguishable, to the concept of statistical decision-making. Since an adversary's goal is to identify whether given privatized $Z$ is from $S$ or $S'$ when $S \sim S'$, we consider the following hypothesis testing problem for a given output of mechanism $M$:

$$H_0 : Z \sim M(S) \quad \text{vs.} \quad H_1 : Z \sim M(S'). \tag{1}$$

The difficulty of this hypothesis testing problem is directly related to the privacy level of the given mechanism. A mechanism is said to satisfy $f$-FDP if the above testing problem has a "trade-off" function $f(\alpha)$, which is the minimum Type II error for a given Type I error no greater than $\alpha \in [0, 1]$. FDP, particularly in the context of Gaussian differential privacy (GDP) (see Section 2), has been found to provide superior privacy control compared to $(\epsilon, \delta)$-DP, with better interpretation.

Another advantage of the FDP framework is its superiority in composition rules, which regards accurately measuring the privacy level when multiple mechanisms are sequentially applied. Many

types of DP, including $(\epsilon, \delta)$-(L)DP, have been criticized for their inefficient composition rules. They tend to overestimate privacy leakage, which, in turn, leads to excessive perturbation of estimations. This is demonstrated in Section 4, in which $\epsilon$-LDP is empirically shown to be inefficient in controlling the privacy of a high-dimensional problem. On the other hand, FDP possesses an effective composition rule that strikes the right balance between privacy protection and estimation accuracy.

## 1.1 Related Works

Since DP was introduced, the trade-off between privacy and statistical utility has been studied in global DP settings as well as local DP settings. Kamath et al. (2022) and Kamath and Ullman (2020) investigated the trade-off under $\epsilon$-DP by studying sample complexity of covariance matrix estimation of Gaussian distribution and univariate mean estimation, respectively. Cai et al. (2021) established minimax optimality under $(\epsilon, \delta)$-DP for high-dimensional mean estimation of sub-Gaussian distributions.

In the local DP setting, analyzing minimax risks under $\epsilon$-LDP for various estimation problems has been a main focus. Duchi et al. (2018) provided bounds of minimax optimal privacy mechanisms for some canonical estimation problems including mean and density estimation and suggested optimal estimators under $\epsilon$-LDP. Other estimation problems that have been investigated under $\epsilon$-LDP include the estimation of functionals of a probability distribution (Rohde and Steinberger, 2020), and discrete distribution estimation (Chhor and Sentenac, 2023). Private minimax risk of nonparametric density estimation has been addressed by Butucea et al. (2020) who observed an elbow effect in $L^r$ risk over Besov spaces and by Li et al. (2022) who considered a data contamination scenario. Some efforts have also been made under $(\epsilon, \delta)$-LDP. Asoodeh et al. (2021) studied minimax risks for mean estimation, while Kroll (2021) considered nonparametric density estimation at fixed points.

Among the existing works, Duchi et al. (2018) and Asoodeh et al. (2021) are more relevant to the present work than others, as they also studied the mean estimation and/or nonparametric density estimation. We show that the minimax optimal rates for $f$-FLDP, local version of $f$-FDP, are equivalent to those for $\epsilon$-LDP, if $f$ satisfies some condition (see Lemma 1). We argue that our result can also be useful for understanding $(\epsilon, \delta)$-LDP, since $f$-FLDP includes $(\epsilon, \delta)$-LDP as a special case.

## 1.2 Our Contributions

We derive the minimax risk bounds under the framework of FLDP as well as Gaussian LDP (GLDP) for two classic statistical problems: univariate mean estimation (Section 3.1) and nonparametric density estimation (Section 3.2). We investigate the private minimax risks using Le Cam's (Theorem 1) and Assouad's bounds (Theorem 3), respectively. To the best of our knowledge, our work is the first to investigate the minimax risk bounds under the FLDP framework.

Our theoretical investigation yields two important results: First, we show that our bounds achieve the same rates under some conditions as those under $\epsilon$-LDP, thereby extending the contraction inequality established by Duchi et al. (2018) under $\epsilon$-LDP to FLDP (Theorem 2). Also, as a special case, we show that the established lower bound becomes tight under GLDP. Second, we demonstrate that $f$-FLDP provides a *continuous* trade-off between privacy and statistical utility, unlike $(\epsilon, \delta)$-LDP (Corollaries 1 and 3). In the view of $(\epsilon, \delta)$-LDP, one can see that $\epsilon$-LDP and non-private settings are continuously related by a quantity $\delta$. However, the existing results do not reflect such continuity with respect to the minimax rate. So far, any slightest privacy measure seems to increase the minimax rate with the increment rate not related to the privacy level. For example, the minimax rate of nonparametric density estimation worsens from $n^{-\frac{2\beta}{2\beta+1}}$ to $n^{-\frac{2\beta}{2\beta+2}}$ under $\epsilon$-LDP, where $\beta$ denotes the degree of smoothness. But the change of rate is irrelevant to $\epsilon$ even though $\epsilon$-LDP becomes non-private if we set $\epsilon \to \infty$.

We also present optimal mechanisms that attain the minimax upper bounds (Corollaries 2 and 4), and evaluate their utilities compared to optimal mechanisms derived under $\epsilon$-LDP. Our experiments show that the proposed minimax optimal estimators under GLDP achieve better utility for the equivalent level of privacy constraints (Section 4).

## 2 Backgrounds

We introduce some notations and review key concepts in differential privacy in this section. Let $M : \mathcal{X}^n \longrightarrow \mathcal{P}(\mathcal{Z})$ be a multivariate randomized mechanism or equivalently, a collection of

randomized mechanisms $M = \{M_i\}_{i=1}^n$ such that $Z = (z_1, \ldots, z_n) := M(x_1, \ldots, x_n)$ where $z_i = M_i(x_1, \ldots, x_n)$ for $i = 1, \ldots, n$. Here, $M$ denotes a locally private mechanism as it privatizes a data set $D \in \mathcal{X}^n$, producing perturbed data with the same size as the original data. Furthermore, $M$ is called a sequential mechanism if $z_i$ depends only on $x_i$ and $z_1, \ldots, z_{i-1}$, so $z_i = M_i(x_i, z_1, \ldots, z_{i-1})$.

## 2.1 Functional Local Differential Privacy

The hypothesis testing aspect of DP expressed in (1) was first discussed by Wasserman and Zhou (2010). Recently Dong et al. (2022) gave a formal treatment by formulating the difficulty of the testing as follows. Let $\phi$ be the rejection rule and $\alpha$ be the level of the test. Then we define the trade-off function[2] as the minimum type II error for a given $\alpha$.

**Definition 2.1** (Trade-off function, Dong et al. (2022)). For two distributions $P, Q \in \mathcal{P}(\mathcal{Z})$ and $\alpha \in [0, 1]$, the trade-off function $T(P, Q) : [0, 1] \longrightarrow [0, 1]$ between $P$ and $Q$ is defined as

$$T(P, Q)(\alpha) = \inf_\phi \left\{ 1 - \int_{\mathcal{Z}} \phi(z) dQ(z) \ \middle| \ \phi : \mathcal{Z} \longrightarrow [0, 1], \int_{\mathcal{Z}} \phi(z) dP(z) \leq \alpha \right\}.$$

It is clear that the larger the trade-off function, the smaller the power, and consequently the more private the mechanism. A randomized mechanism is said to satisfy $f$-FDP if the corresponding trade-off function is at least $f$ for all $\alpha \in [0, 1]$.

**Definition 2.2** (Functional differential privacy, FDP), Dong et al. (2022)). Let $f : [0, 1] \longrightarrow [0, 1]$ be a trade-off function for some distributions $P$ and $Q$. A given mechanism $M$ is $f$-FDP if

$$T(M(S), M(S'))(\alpha) \geq f(\alpha)$$

for every $S \sim S' \subset \mathcal{X}^n$ and $\alpha \in [0, 1]$.

The following proposition shows that $(\epsilon, \delta)$-DP defined in (1.1) is a special case of $f$-FDP.

**Proposition 1** (Dong et al. (2022)). *A convex conjugate of $f$ is defined as $\delta_f(y) = \sup_{x \in [0,1]} 1 - yx - f(x)$. Then a mechanism $M$ satisfies an $f$-FDP iff it is $(\epsilon, \delta_f(e^\epsilon))$-DP for every $\epsilon \geq 0$.*

According to Proposition 1, all information regarding the privacy of the mechanism from the perspective of $(\epsilon, \delta)$-DP is contained in FDP. It also indicates that the privacy characterization of a mechanism may require more than just two numbers $\epsilon$ and $\delta$.

A useful subclass of FDP is the Gaussian differential privacy (GDP), which has the trade-off function $G_\mu = T(N(0, 1), N(\mu, 1))$. That is, the hypothesis testing problem in (1) compares two normal distributions with respective means 0 and $\mu$ with unit variance. It is known that $G_\mu(x) = F(F^{-1}(1 - x) - \mu)$ where $F$ is the cumulative distribution function of the standard normal distribution.

**Definition 2.3** (Gaussian differential privacy, GDP). A mechanism $M$ is $\mu$-GDP if it is $G_\mu$-FDP.

In addition to its intuitive interpretation, $\mu$-GDP also possesses appealing asymptotic properties regarding compositions of different DP mechanisms. It has been shown by Dong et al. (2022) that the trade-off function of a composition of mechanisms converges to $G_\mu$ under some mild conditions. Due to these desirable features, it is suggested to be used as a standard for the comparison of different DP methods.

In this work, we focus on the utility of estimation under local $\mu$-GDP, i.e., $\mu$-GLDP. For all DP concepts introduced so far, their corresponding local versions are naturally defined by a locally private sequential mechanism $M = \{M_i\}$, $i = 1, \ldots n$, that privatizes the $i$th observation, respectively. For example, $f$-FLDP and $\mu$-GLDP are characterized by the following:

$$T(M_i(x), M_i(x'))(\alpha) \geq f(\alpha) \ (\text{or } G_\mu(\alpha))$$

for every $x, x' \in \mathcal{X}$ and $i = 1, 2, \ldots, n$. Some exemplary trade-off functions of $\mu$-GLDP as well as those of $\epsilon$-LDP are displayed in Fig. 1a.

---

[2]not to be confused with the *trade-off* between privacy and statistical utility.

## 2.2 Private Minimax Risk

Consider an estimation problem in a private setting. Suppose $n$ i.i.d. observations $X_1, \ldots, X_n \in \mathcal{X}$ are drawn from a distribution $P$ in some family of distributions $\mathcal{P} \subset \mathcal{P}(\mathcal{X})$. The data are privatized into $(Z_1, \ldots, Z_n) = M(X_1, \ldots, X_n)$ where $M$ is an $f$-FLDP mechanism. Let $\theta = \theta(\mathcal{P}) \in \Theta$ be a parameter of interest where $\Theta$ is the parameter space and $\hat{\theta} : \mathcal{X}^n \longrightarrow \Theta$ be an estimator. For a metric $\rho : \Theta^2 \longrightarrow \mathbb{R}_{\geq 0}$ and an increasing function $\Phi : \mathbb{R}_{\geq 0} \longrightarrow \mathbb{R}_{\geq 0}$, use $\Phi \circ \rho$ as the loss function, we define the private minimax risk:

**Definition 2.4** (Private minimax risk)**.**

$$\mathcal{R}_n(\theta(\mathcal{P}), \Phi \circ \rho, M_f) := \inf_{M \in M_f} \inf_{\hat{\theta}} \sup_{P \in \mathcal{P}} \mathbb{E}_P \left[ \Phi \circ \rho \left( \hat{\theta} \left( M(X_1, \ldots, X_n) \right), \theta(P) \right) \right].$$

Here, $M_f$ is a family of $f$-FLDP mechanism defined over $\mathcal{X}^n$. As long as privacy is guaranteed at a certain level (i.e., given $f$), one wishes to find an optimal privatizing mechanism $M$ for a given estimation problem (i.e., $\theta$) that yields uniformly optimal loss over every distribution in $\mathcal{P}$. One can also use the minimax risk not only to find the optimal achievable estimation risk but to judge the efficiency of the estimator under the given privacy level $f$-FLDP.

# 3 Minimax Risk Analysis Under FLDP

Minimax risks of classical, non-private estimation problems have been addressed by numerous approaches including Le Cam, Fano, and Assouad methods, among others (Tsybakov, 2009). In this section, we take Le Cam's and Assouad's approaches and apply them to mean and nonparametric density estimation, respectively, under $f$-FLDP. All proofs are presented in the supplementary material.

## 3.1 Le Cam's Bound on Mean Estimation

Le Cam's method involves reducing an estimation problem to a two-point hypothesis testing problem, in which the minimax risk is bounded by calculating the worst-case risk for only two distributions, $P_1$ and $P_2$, selected from $\mathcal{P}$, as defined in Section 2.2.

We first state the private Le Cam's lower bound under $f$-FLDP in the theorem below. The following quantity represents the contraction factor of the effective sample size due to the privatization:

$$c_{f, \kappa} = 2\kappa^\kappa (1 - \kappa)^{1-\kappa} \int_0^\infty (1 + \kappa) t^{\kappa - 1} \delta_f(t) dt,$$

where $0 \leq \kappa \leq 1$ and $\delta_f(t)$ is the convex conjugate of $f$, as defined in Proposition 1.

**Theorem 1** (Private Le Cam's bound)**.** *For given $0 \leq \kappa \leq 1$ and a trade-off function $f$, if $\int_0^\infty t^{\kappa-1}\delta_f(t)dt$ is finite and $\rho(\theta(P_1), \theta(P_2)) \geq 2\eta > 0$ for two distributions $P_1, P_2 \in \mathcal{P}$, then*

$$\mathcal{R}_n(\theta(\mathcal{P}), \Phi \circ \rho, M_f) \geq \frac{\Phi(\eta)}{2} \left[ 1 - \sqrt{n c_{f,\kappa} \frac{\|P_1 - P_2\|_{TV}^{1+\kappa}}{(1 - \|P_1 - P_2\|_{TV})^\kappa}} \right],$$

*where $\|\cdot\|_{TV}$ denotes total variation.*

The minimax lower bound on the right-hand side can serve as an indicator of the trade-off between privacy and utility. Note that $c_{f,\kappa} \leq c_{g,\kappa}$ if $f(t) \geq g(t)$ for all $t \in [0, 1]$, which implies higher privacy would yield greater contraction of the effective sample size. That is, the minimax lower bound on the right-hand side would become lower under a loosened privacy constraint, which coincides with the intuition that less privacy would enhance the utility.

Next, we use Le Cam's method to obtain a lower bound of minimax risk for univariate mean estimation with bounded moments under the squared loss. For $k > 1$, define a family of distributions with bounded $k$th moment $\mathcal{P}_k = \{P \in \mathcal{P}(\mathbb{R}) \mid |\mathbb{E}_P[X]| \leq 1, \mathbb{E}_P[|X|^k] \leq 1\}$. The parameter of interest is $\theta(P) = \mathbb{E}_P[X]$ with loss function $\Phi \circ \rho$ where $\rho(\theta_1, \theta_2) = |\theta_1 - \theta_2|$ and $\Phi(t) = t^2$. Applying Theorem 1, we obtain the following result.

**Corollary 1** (Univariate mean estimation). *For given $0 \leq \kappa \leq 1$ and a trade-off function $f$, if $\int_0^\infty t^{\kappa-1}\delta_f(t)\,dt$ is finite, then*

$$\mathcal{R}_n\left(\theta(\mathcal{P}_k), |\cdot|^2, M_f\right) \geq c_0\left(nc_{f,\kappa}\right)^{-\frac{1}{1+\kappa}\frac{2(k-1)}{k}},$$

*for $n > c_1$ where $c_0$ only depends on $k$ and $\kappa$, and $c_1$ depends on $c_{f,\kappa}$.*

Note that the minimax risk for mean estimation is known to be $O\left(n^{-\min\{1, 2-\frac{2}{k}\}}\right)$ in a non-private setting and $O\left((n\epsilon^2)^{-\frac{k-1}{k}}\right)$ under $\epsilon$-LDP (Duchi et al., 2018). Corollary 1 implies that the minimax lower bound continuously reaches the two scenarios. Specifically, the minimax risk bound of $\kappa = 0$ corresponds to that of the non-private setting, and the minimax risk bound of $\kappa = 1$ corresponds to that of the $\epsilon$-LDP case. Moreover, the fact that for any trade-off function $f$, there exists $f_\epsilon$ such that $f_\epsilon$-FLDP is equivalent with $\epsilon$-LDP implies that the minimax rate of mean estimation under $f$-FLDP is $O\left(n^{-\frac{k-1}{k}}\right)$ when $c_{f,1}$ is finite. In the following lemma, we further identify which trade-off function $f$ would yield the private minimax risk bound of $f$-FLDP equivalent to that of $\epsilon$-LDP.

**Lemma 1.** *If a given trade-off function $f(x) \geq 1 - c_0 x^{c_1}$ for some $c_0 > 0$ and $c_1 \in (0.5, 1)$ on $[0,1]$, or $\delta_f(x) \leq c_3 x^{-1-c_2}$ for $x > x_0$ and some $c_3, c_2, x_0 > 0$, then $\int_0^\infty \delta_f(t)\,dt$ is finite.*

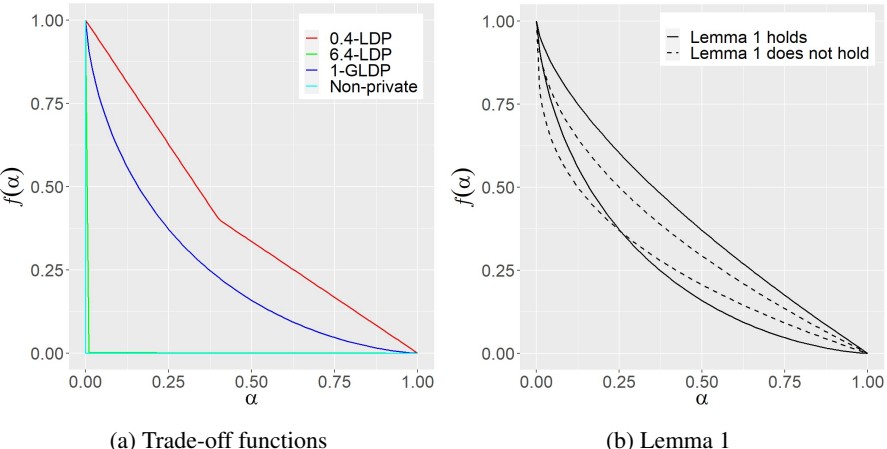

(a) Trade-off functions          (b) Lemma 1

Figure 1: (a) Trade-off functions of $\epsilon$-LDP with $\epsilon = 0.4$ (red), 6.4 (green), $\mu$-GLDP with $\mu = 1$ (blue) and non-private setting (cyan). (b) Example of trade-off functions (not) satisfying Lemma 1.

Under the condition of Lemma 1, the minimax risk of univariate mean estimation under $f$-FLDP is $O(n^{-\frac{k-1}{k}})$ by Corollary 1, which is the same rate for $\epsilon$-LDP. Here, we note that the assumption of Lemma 1 is quite general. Fig. 1b shows some examples of trade-off functions satisfying Lemma 1. It can be seen that Lemma 1 requires the slope at $x = 0$ of the trade-off function should not be too steep. Nonetheless, note that there exists a trade-off function requiring $\kappa < 1$ in order to make $c_{f,\kappa}$ finite. For example, $f(x) = 1 - x^{\frac{\kappa}{1+\kappa}}$ is a trade-off function with $\delta_f(t) = \frac{\kappa^\kappa}{(1+\kappa)^{1+\kappa}}t^{-\kappa}$ for $t > \frac{\kappa}{1+\kappa}$. Therefore, $\int t^{\kappa-1}\delta_f(t)dt$ does not converge, and neither does $c_{f,\kappa}$. Thus, there exist some mechanisms that Corollary 1 cannot guarantee the minimax rate equivalent with the optimal rate under $\epsilon$-LDP. In other words, they may enjoy better minimax rates. In the supplementary material, it is shown that $G_\mu$ satisfies Lemma 1 for any $\mu > 0$.

Combining it with Corollary 1, we can bound the minimax risk of univariate mean estimation under $\mu$-GLDP. Denote $M_\mu$ for a family of $\mu$-GLDP mechanisms.

**Corollary 2.** *The minimax risk of univariate mean estimation under $\mu$-GLDP mechanism is bounded as follows:*

$$c_0\left(ne^{\frac{1}{2}\mu^2}\right)^{-\left(1-\frac{1}{k}\right)} \leq \mathcal{R}_n\left(\theta\left(\mathcal{P}_k, |\cdot|_2^2, M_\mu\right)\right) \leq c_1(n\mu^2(4+\mu^2)^{-1})^{-\left(1-\frac{1}{k}\right)},$$

*for $n > c_2$ where $c_0, c_1 > 0$ are constants depending only on $k$, and $c_2$ is a constant depending only on $\mu$ and $k$.*

We present the following $\mu$-GLDP algorithm that can be shown to be minimax optimal, as it achieves the rate of $O(n^{-\frac{k-1}{k}})$. See the supplementary material for a proof and empirical comparison of other LDP mechanisms including the minimax optimal $\epsilon$-LDP mechanism in Duchi et al. (2018).

1. $T = \left[ n \left( 1 + \frac{4}{\mu^2} \right)^{-1} \right]^{\frac{1}{2k}}$.

2. $M_i(X_i) = \max\{-T, \min\{X_i, T\}\} + \epsilon_i$ where $\epsilon_i \sim N\left(0, \frac{4T^2}{\mu^2}\right)$.

3. $\hat{\theta}\left(M(X_1, \ldots, X_n)\right) := \frac{1}{n} \sum_{i=1}^n M_i(X_i)$.

The fact that $\mu$-GLDP and $\epsilon$-LDP have the same minimax rate might be unexpected to some, especially considering that the former is a more relaxed concept than the latter. In the following lemma, we present an interesting inequality that can shed light on their connection.

**Lemma 2.** *Let $M : \mathcal{X} \longrightarrow \mathcal{P}(\mathcal{Z})$ be a locally private mechanism taking only one data for its input. For $P_1, P_2 \in \mathcal{P}(\mathcal{X})$, denote $m_i$ for density of $M(P_i)$ for $i = 1, 2$. If $M$ is $f$-FLDP, then for $a \geq 2$,*

$$\mathbb{P}_{m_2}\left(\frac{m_1(Z)}{m_2(Z)} > a\right) \leq \delta_f(a - 1).$$

Lemma 2 implies that, for sufficiently small $\delta_f(e^\epsilon - 1)$, an $f$-FLDP mechanism behaves like $\epsilon$-LDP with high probability $(1 - \delta_f(e^\epsilon - 1))$. Moreover, this lemma provides a probabilistic interpretation for $(\epsilon, \delta)$-LDP: the probability that the likelihood ratio between output distributions exceeds $e^\epsilon + 1$ is bounded by $\delta$. In the DP literature (Mironov, 2017; Meiser, 2018), $(\epsilon, \delta)$-DP is often informally explained as "an $\epsilon$-DP with probability $1 - \delta$," which can cause confusion. In this regard, here we provide an accurate statement specifying the respective roles of $\epsilon$ and $\delta$.

In addition, the $\mu$-GLDP mechanism possesses a similar contraction ability to $\epsilon$-LDP. The term *contraction* of a mechanism is generally used to describe the reduction of distance measures between distributions that is altered by a mechanism. In Duchi et al. (2018), it was shown that $D_{kl}\left(M(P_1)\|M(P_2)\right)$, the Kullback-Leibler (K-L) divergence between privatized distributions under $\epsilon$-LDP, is bounded by $2(e^\epsilon - 1)^2 \|P_1 - P_2\|_{TV}^2$. We establish a similar bound under $f$-FLDP. Following the notations in Lemma 2, the following theorem describes the contraction of $f$-FLDP mechanism in terms of K-L divergence.

**Theorem 2.** *For given $0 \leq \kappa \leq 1$ and a trade-off function $f$, if $\int_0^\infty t^{\kappa-1}\delta_f(t)dt$ is finite, then*

$$D_{kl}^{sy}(m_1\|m_2) := D_{kl}(m_1\|m_2) + D_{kl}(m_2\|m_1) \leq c_{f,\kappa} \frac{\|P_1 - P_2\|_{TV}^{1+\kappa}}{(1 - \|P_1 - P_2\|_{TV})^\kappa}$$

*holds for any $f$-FLDP mechanism $M : \mathcal{X} \longrightarrow \mathcal{P}(\mathcal{Z})$ and distributions $P_1$ and $P_2$ over $\mathcal{X}$.*

In the case of $\mu$-GLDP, the bound comes down to $O\left(\|P_1 - P_2\|_{TV}^2\right)$ since $\int_0^\infty t^{\kappa-1}\delta_f(t)dt$ is finite for $\kappa = 1$. Because $f$-FLDP includes $\epsilon$-LDP, our result is more general than Duchi et al. (2018) and further gives an insight into why $\epsilon$-LDP and $\mu$-GLDP have indistinguishable privatizing power. Our analysis also improves the bound found by Asoodeh et al. (2021), which is expressed as a constant multiple of $D_{kl}(P_1\|P_2)$ thus could be unbounded for point distributions. Our result bounds the divergence by total variation between $P_1$ and $P_2$, which is finite for any two arbitrary distributions. See the supplementary material for an elaborated illustration.

### 3.2 Assouad's Bound for Nonparametric Density Estimation

This section aims to derive the minimax risks of private nonparametric density estimation under $f$-FLDP and $\mu$-GLDP. While Le Cam's method is effective for many problems, it may not be suitable for high-dimensional structured problems. In such cases, Assouad's method offers a solution by reformulating the estimation problem as a multiple-binary hypothesis testing problem. Given the parameter space $\Theta$, there exists a map $V : \Theta \longrightarrow \{-1, +1\}^d$ and a family of distributions $\{P_v\}_{v \in \{-1, +1\}^d}$ for $d \in \mathbb{N}$ such that

$$\Phi(\rho(\theta, \theta(P_v))) \geq 2\eta \sum_{j=1}^d \mathbf{1}\{[V(\theta)]_j \neq v_j\}$$

for every $v \in \{-1, +1\}^d$. We say $\{P_v\}_{v \in \{-1,+1\}^d}$ induces a $2\eta$-Hamming separation under loss $\Phi \circ \rho$. Assouad's method establishes a lower bound of minimax risk by assuming not the worst case but a randomly selected case in $\{P_v\}_{v \in \{-1,+1\}^d}$. We present the Assouad's bound under $f$-FLDP mechanisms.

**Theorem 3** (Private Assouad's bound). *For given $0 \leq \kappa \leq 1$ and a trade-off function $f$, if $\int_0^\infty t^{\kappa-1} \delta_f(t)\, dt$ is finite and a set of distributions $\{P_v\}_{v \in \{-1,+1\}^d} \subset \mathcal{P}$ induces $2\eta$-Hamming separation under $\Phi \circ \rho$, then*

$$\mathcal{R}_n\left(\theta(\mathcal{P}), \Phi \circ \rho, M_f\right) \geq d\eta \left[ 1 - \sqrt{\frac{n c_{f,\kappa}}{d} \sum_{j=1}^d \frac{\|P_{+j} - P_{-j}\|_{TV}^{1+\kappa}}{(1 - \|P_{+j} - P_{-j}\|_{TV})^\kappa}} \right] \qquad (2)$$

*where $P_{\pm j} = \frac{1}{2^{d-1}} \sum_{v:v_j=\pm 1} P_v$.*

We can use Theorem 3 to obtain the lower bound of minimax risk of private nonparametric density estimation. The true density is assumed to be in the elliptical Sobolev space, as defined below:

**Definition 3.1** (Elliptical Sobolev space). *For a given orthonormal basis $\{\phi_j\}_{j=1}^\infty$ of $L^2([0,1])$, smoothness parameter $\beta > \frac{1}{2}$, and radius $r > 0$, the elliptical Sobolev space is the following set:*

$$\mathcal{F}_\beta[r] = \left\{ h \in L^2[0,1] \middle| h = \sum_{j=1}^\infty \theta_j \phi_j, \sum_{j=1}^\infty j^{2\beta} \theta_j^2 \leq r^2 \right\}.$$

If we use the trigonometric basis, then $\mathcal{F}_\beta$ represents a set of distributions with the bounded norm of $\beta$-derivatives with coinciding derivatives of degree$< \beta$ at endpoints of $[0,1]$ (Tsybakov, 2009). Applying Theorem 3, we can obtain the following lower bound.

**Corollary 3** (Nonparametric density estimation). *For given $0 \leq \kappa \leq 1$ and a trade-off function $f$, if $\int_0^\infty t^{\kappa-1} \delta_f(t) dt$ is finite, then*

$$\mathcal{R}_n\left(\theta(\mathcal{F}_\beta[r]), \|\cdot\|_2^2, M_f\right) \geq r^{\frac{2}{\beta+1}} c_1 (n c_{f,\kappa})^{-\frac{2\beta}{(\beta+1)(1+\kappa)}}$$

*for $n > c_2$ where $c_1, c_2 > 0$ are constant depends on $\beta$.*

It is known that the minimax risk in the non-private setting is $O\left(n^{-\frac{2\beta}{2\beta+1}}\right)$ (Tsybakov, 2009) and $O\left((n\epsilon^2)^{-\frac{2\beta}{2\beta+2}}\right)$ under $\epsilon$-LDP (Duchi et al., 2018). Thus according to Corollary 3, the minimax rate of $f$-FLDP with $\kappa = 1$ is the same as that of $\epsilon$-LDP. As in Section 3.1, we can obtain the bounds of minimax risk of nonparametric density estimation under $\mu$-GLDP, using Lemma 1 .

**Corollary 4.** *The minimax risk of nonparametric density estimation under $\mu$-GLDP is bounded as follows:*

$$c_0 r^{\frac{2}{\beta+1}} \left(n e^{\frac{1}{2}\mu^2}\right)^{-\frac{2\beta}{2\beta+2}} \leq \mathcal{R}_n\left(\theta(\mathcal{F}_\beta[r]), \|\cdot\|_2^2, M_\mu\right) \leq c_1 r^{\frac{2}{\beta+1}} (n\mu^2)^{-\frac{2\beta}{2\beta+2}}$$

*for $n > c_2$ where $c_0, c_1 > 0$ are constants depending only on $\beta$, and $c_2$ is a constant depending only on $\mu$ and $\beta$.*

Note that the minimax rate in Corollary 4 is essentially the same as that of $\epsilon$-LDP. However, under $\mu$-GLDP, one can find the optimal mechanism in a more straightforward way. This is in contrast to the discussion in Duchi et al. (2018) that the Laplacian mechanism, the canonical mechanism under $\epsilon$-LDP, cannot achieve the optimal minimax rate. They instead proposed a complicated mechanism to achieve optimality. Our minimax optimal mechanism and density estimator are presented below:

Suppose that $n$ i.i.d. data $X_1, X_2, \ldots, X_n$ drawn from $h \in \mathcal{F}_\beta[r]$ are available.

1. Set $d = \lfloor (0.5 n \mu^2 r^2)^{\frac{1}{2\beta+2}} \rfloor$ where $\lfloor x \rfloor$ is the largest integer not exceeding $x$.

2. $M_i(X_i) := (\phi_1(X_i), \ldots, \phi_d(X_i))^T + \epsilon$ where $\epsilon \sim N(\mathbf{0}, \frac{2d}{\mu^2}\mathbf{I}_{d \times d})$.

3. Define a function $\hat{h}_i : [0,1] \longrightarrow \mathbb{R}$ as $\hat{h}_i := \sum_{j=1}^d [M_i(X_i)]_j \phi_j$.

4. Repeat 2 and 3 for $i = 1, \ldots, n$.

5. $\hat{h} = \frac{1}{n} \sum_{i=1}^n \hat{h}_i$ is the obtained estimator.

# 4 Empirical Comparison of Minimax Optimal Mechanisms

In this section, we compare the optimal mechanisms for nonparametric density estimation under $\epsilon$-LDP, $\mu$-GLDP, and non-private mechanisms. The simulation on univariate mean estimation is provided in the supplementary material. Our proposed mechanism in Section 3.2 under $\mu$-GLDP is compared with the mechanism in Duchi et al. (2018) and the non-private mechanism, which is obtained by using $d = \lfloor (nr^2)^{\frac{1}{2\beta+1}} \rfloor$ with $\epsilon = 0$ in our $\mu$-GLDP algorithm.

The true underlying density is given as the density of $\mathrm{Beta}(5,5)$ distribution.[3] We set $\epsilon \in \{0.4, 0.8, 1.6, 3.2, 6.4\}$ for $\epsilon$-LDP and $\mu \in \{0.5, 1\}$ for $\mu$-GLDP. We vary the sample size from $n = 100$ to $n = 2000$. Estimation errors are calculated by integrated mean squared errors. We repeat the experiment 100 times for each mechanism.

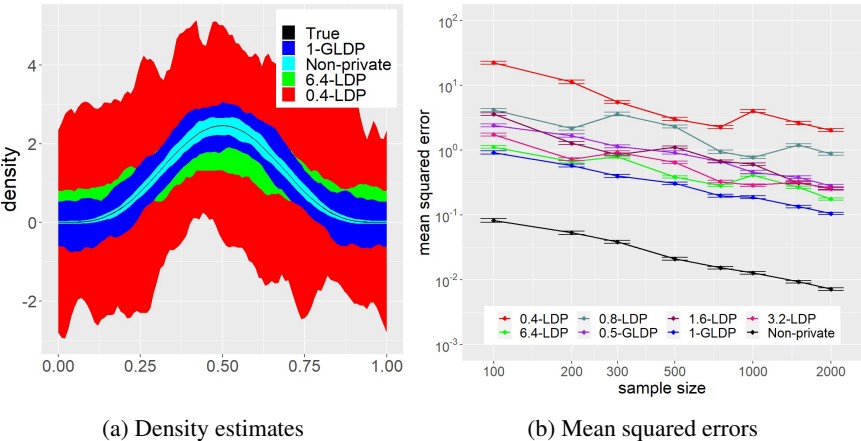

(a) Density estimates          (b) Mean squared errors

Figure 2: (a) 95% confidence intervals calculated from private density estimates over 100 repetitions for three LDP and non-private mechanisms. (b) The mean squared errors with one-standard errors of density estimates for various LDP mechanisms and sample sizes.

The trade-off functions in Fig. 1a show that 6.4-LDP is very close to the non-private setting and 1-GLDP is between 0.4-LDP and 6.4-LDP almost everywhere in terms of privacy levels. However, Fig. 2a, displaying the 95% point-wise confidence intervals obtained from estimated densities by different methods, shows that estimates under 1-GLDP are more in agreement with the true density than 6.4-LDP. The performance under 0.4-LDP is much worse than the others because it is the most private mechanism. The superiority of 1-GLDP can also be seen in terms of mean squared errors. In Fig. 2b, 1-GLDP performs better than all other $\epsilon$-LDPs.

The utility of a private mechanism in this density estimation essentially depends on how well it approximates the estimates of Fourier coefficients that would be obtained from the original data. This task requires an efficient mechanism to estimate a multi-dimensional coefficient vector for each observation. The optimal mechanism for this task under $\epsilon$-LDP transforms a vector of non-private Fourier coefficient estimates into a binomial vector, which limits the support in order to reduce variance and achieve privacy. However, due to the inherent variance introduced by discretization, the mean squared error is bounded by a constant multiple of $\left(\frac{e^\epsilon+1}{e^\epsilon-1}\right)^2$. Explicitly, the optimal error bound of the $\epsilon$-LDP density estimation mechanism in Duchi et al. (2018) can be derived as follows:

$$\mathcal{R}_\epsilon \le (\beta+1)\left(\frac{\beta}{\sqrt{2\pi e}}n\left(\frac{e^\epsilon+1}{e^\epsilon-1}\right)^{-2}\right)^{-\frac{2\beta}{2\beta+2}} r^{\frac{2}{\beta+1}}$$

where $\mathcal{R}_\epsilon$ is the expected error obtained from the $\epsilon$-LDP density estimation mechanism in Duchi et al. (2018). Hence, even with a lenient privacy constraint (i.e., a large $\epsilon$), the estimation error may not decrease significantly. In contrast, our $\mu$-GLDP optimal mechanism generates an estimator by adding noise with decreasing variance as $\mu$ tends to infinity. Again, the explicit optimal error bound is given

---

[3]We use $\beta = 3$ and $r = 408.8979$ for the proposed mechanism in Section 3.2

as

$$\mathcal{R}_\mu \le (\beta + 1)\left(0.5\beta n\mu^2\right)^{-\frac{2\beta}{2\beta+2}} r^{\frac{2}{\beta+1}} + O\left(n^{-\frac{2\beta+1}{2\beta+2}}\right)$$

where $\mathcal{R}_\mu$ is the expected error obtained from our $\mu$-GLDP density estimation mechanism. The comparison of the coefficients of $n^{-\frac{2\beta}{2\beta+2}}$, which can be interpreted as the cost of privacy, gives the asymptotic error ratio:

$$\frac{\mathcal{R}_\mu}{\mathcal{R}_\epsilon} \approx \left(\sqrt{\frac{\pi e}{2}}\mu^2\left(\frac{e^\epsilon + 1}{e^\epsilon - 1}\right)^2\right)^{-\frac{2\beta}{2\beta+2}},$$

which exceeds 1 when $\mu = 1$. Therefore, the estimation under $\epsilon$-LDP is likely to produce higher errors than the estimation under 1-GLDP for any $\epsilon$. This is also evidenced by the result in Fig. 2a. The fact that private estimation error under $\epsilon$-LDP does not approach non-private estimation error implies that $\epsilon$-LDP adds more noise than necessary to high-dimensional data compared to $\mu$-GLDP. This highlights the limitation of $\epsilon$-LDP and the advantage of $\mu$-GLDP in achieving privacy for multi-dimensional data, which can be practically beneficial for addressing other private estimation problems.

## 5   Conclusion

The present work on $f$-FLDP establishes the private Le Cam and Assouad bounds reflecting the continuous nature of privacy constraints. As the trade-off function $f$ becomes more (less) private, the supremum value of $\kappa$ satisfying the conditions of Corollaries 1 and 3 increases (decreases), leading to the $\epsilon$-LDP (non-private, respectively) minimax rate in terms of $n$. This is in contrast to the somewhat counter-intuitive minimax lower bounds derived under $\epsilon$-LDP, whose utility never achieves that of non-private estimation even as $\epsilon$ approaches infinity. Our simulation study in Section 4 shows that $\mu$-GLDP can offer similar or better privacy level than $\epsilon$-LDP, while achieving higher accuracy in private estimation. This suggests that $\mu$-GLDP may be a better option for achieving both privacy and accurate estimation. The results on the contraction inequality reported in Theorem 2 and in the supplementary material provide additional support for this claim based on the privatizing characteristics of the mechanisms. In summary, this work shows that $\mu$-GLDP is theoretically comparable in terms of both utility and privatizing ability while enjoying the better practical performance. This supports the suggestion made by Dong et al. (2022) that we use G(L)DP as a reliable standard for (L)DP both theoretically and empirically.

Possible limitations of this work are as follows: Although we establish lower bounds that connect privacy and utility in a continuous manner through $\kappa \in [0, 1]$, the suggested optimal mechanisms only achieve the minimax rates for $\kappa = 1$. It will be desirable to identify private mechanisms and estimators that can achieve minimax rates between the non-private and $\epsilon$-LDP minimax rates, if such mechanisms exist. Additionally, the condition in Theorems 1 and 3 does not hold for all possible trade-off functions. Also, the bounds in Corollaries 1 and 3 require $\int t^{\kappa-1}\delta_f(t)dt$ to be finite, which restricts the potential application of the theory, even though most commonly used privatization mechanisms, such as the Gaussian and Laplace mechanisms, satisfy this condition. Nevertheless, these bounds only coincide with the true minimax rates for only limited categories of privacy. Hence, we plan to explore whether there exists an optimal mechanism that enjoys a better minimax rate for mean and density estimation. Additionally, our theoretical minimax bound for estimations under $\mu$-GLDP is tight in terms of the sample size $n$, but it notably lacks tightness with respect to the privacy parameter $\mu$. Our lower bound converges to a finite value when $\mu$ goes to $0$ incompatible with the upper bound. Hence, further efforts are required to establish a rigorous minimax rate in terms of privacy constraints for local FDP. Finally, our results in Section 3.1 are restricted to univariate mean estimation, and extending them to high-dimensional estimation under non-$\epsilon$-LDP would be a non-trivial task.

### Acknowledgement

This work was supported by the Institute of Information & Communications Technology Planning & Evaluation (IITP) grant funded by the Korean government (MSIT) (No.2022-0-00937, Solving the problem of increasing the usability and usefulness of synthetic data algorithms for statistical data.) The work of Jeongyoun Ahn was partially supported by the National Research Foundation of Korea (NRF-2021R1A2C1093526, RS-2023-00218231). The work of Cheolwoo Park was partially supported by the National Research Foundation of Korea (NRF-2021R1A2C1092925).

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
