# OpenReview forum: "Minimax Risks and Optimal Procedures for Estimation under Functional Local Differential Privacy"
_NeurIPS.cc/2023/Conference — NeurIPS 2023 poster_

### Official Review · Reviewer_H56L · 2023-07-05

**Soundness:** 3 good
**Presentation:** 3 good
**Contribution:** 2 fair
**Rating:** 6
**Confidence:** 3

**Summary:**

The authors consider the a local version of functional DP / Gaussian DP, and establish minimax rates of convergence for mean estimation and density estimation under this privacy paradigm.  They highlight how functional/Gaussian DP is more conducive to LDP than approximate DP given it’s tight compositional properties.   They demonstrate how properties of the tradeoff function play a critical role in the minimax risk.

**Strengths:**

This is a timely work given the recent attention GDP has received.  Understanding minimax rates for LDP in such a setting seems like an important contribution.  The mathematical results are thorough, and translating properties of the tradeoff function into minimax rates is interesting.

**Weaknesses:**

A seemingly glaring problem with results such as Corollary 2 is the privacy level, $\mu$, is not made explicit in the rate.  However the constants in the theorem do depend on $\mu$.  The results of Duchi et al make the dependence on epsilon explicit, and it is quite substantial (i.e. taking substantially smaller values of \epsilon will substantially reduce the effective sample size).   The authors then discuss how surprising it is that $\epsilon$ LDP and $\mu$ GLDP would have the same minimax rate, but the authors didn’t show this since their rate does not include $\mu$.  I find this unfortunate, as otherwise the paper is quite interesting, but the final comparison with $\epsilon$ LDP falls flat and unfinished.

**Questions:**

See weaknesses.

---

> ### Author Rebuttal · Authors · 2023-08-09
>
> **Q**: A seemingly glaring problem with results such as Corollary 2 is the privacy level, $\mu$, is not made explicit in the rate. However the constants in the theorem do depend on $\mu$. The results of Duchi et al make the dependence on epsilon explicit, and it is quite substantial (i.e. taking substantially smaller values of epsilon will substantially reduce the effective sample size). The authors then discuss how surprising it is that LDP and GLDP would have the same minimax rate, but the authors didn’t show this since their rate does not include $\mu$. I find this unfortunate, as otherwise the paper is quite interesting, but the final comparison with LDP falls flat and unfinished.
>
> **A**: We appreciate the reviewer's thoughtful comments and insightful feedback. As demonstrated in the paper, both LDP and GLDP share the same minimax rate concerning $n$. However, their equivalence does not extend to privacy constraints in our theoretical results. In response to the reviewer's suggestion, we have undertaken additional calculations to determine the minimax rate while incorporating privacy constraints. The derived bounds are as follows:
> \begin{equation*}
> O\left(\left(ne^{\mu^2}\right)^{-\frac{2(k-1)}{2k}\textbf{or}-\frac{2\beta}{2\beta+2}}\right)\leq\mathcal{R}\leq O\left(\left(n\mu^2\right)^{-\frac{2(k-1)}{2k}\textbf{or}-\frac{2\beta}{2\beta+2}}\right)
> \end{equation*}
> for both univariate mean estimation or nonparametric density estimation.
> This analysis highlights that our approach does not yield a unified minimax rate in terms of $\mu$. Furthermore, the observation that $e^{\mu^2}\longrightarrow1$ as $\mu\longrightarrow0$ contrasts with the behavior of the lower bounds presented in Duchi et al. (2018), which tend towards infinity as $\epsilon\longrightarrow0$.  Hence, further efforts are required to establish a rigorous minimax rate in terms of privacy constraints for local FDP.

---

> > ### Comment · Reviewer_H56L · 2023-08-15
> >
> > Thank you for the reply, I think this is an interesting point.

---

### Official Review · Reviewer_fz6L · 2023-07-07

**Soundness:** 3 good
**Presentation:** 4 excellent
**Contribution:** 3 good
**Rating:** 7
**Confidence:** 3

**Summary:**

The authors study the problems of mean estimation and density estimation under functional local differential privacy (FLDP). In particular, they are interested in deriving minimax (rate-) optimal estimation procedures and privacy mechanisms for these problems. Their results include analogues to Le Cam's bound and Assouad's bound for the FLDP setting. They then specialize these results to the previously mentioned problems and derive minimax lower bounds as corollaries. They also provide algorithms which match the rates specified by their lower bounds. Finally, numerical experiments confirm empirically that their methods achieve better privacy/utility tradeoffs than existing methods which enforce LDP.

**Strengths:**

This work is the first to address minimax estimation rates for functional local DP. Functional DP was introduced recently and has received a great deal of attention, and understanding its properties and advantages is of great interest to the community. As such, this paper offers a relevant and novel contribution. I did not check all of the proofs, but those I did were technically sound.

The results in the paper are quite extensive. The authors not only derive the minimax rate for two fundamental estimation problems (univariate mean estimation and density estimation), but also give algorithms which match the minimax rate. In addition to these strong theoretical results, their algorithms also offer practical improvement over existing alternatives, as can be seen from their empirical evaluation.

The paper is also very well written. I found the background section to be a very helpful introduction to both functional local DP and private minimax estimators for a non-expert. The authors also give helpful intuition and interpretations for many of their technical results, which make this highly technical paper much easier to parse.

**Weaknesses:**

While the authors did a good job providing interpretation for many of their results to make the paper accessible to non-experts, some results still seemed opaque to me. For instance, the instantiation of the Le Cam's and Assouad's bounds provided some intuition for how these results could be useful, but I did not grasp the intuition for either of the more general results (Theorems 1 & 3). In particular, it seems like for large n, the lower bound in both of these theorems will actually be negative and therefore vacuous. Adding some further interpretation of the general bounds would be helpful.

The authors also make a point that their results smoothly interpret between private and non-private regimes, a feature which is lacking from existing analyses. This is of course a strength of the paper (mentioned above), but I did not understand the connection with $\kappa$ in the general lower bounds. In particular, I see that the minimax optimal rates for mean estimation are recovered for $\kappa = 0,1$, but it was not clear to me why these values of $\kappa$ correspond to private/non-private settings. A more thorough explanation of this (probably in terms of the relationship between $\kappa$ and the tradeoff function $f$) would be helpful.

Lastly, the experimental section is fairly sparse. Since this is primarily theoretical paper, I think it is a fairly minor point.

**Questions:**

1. Can the authors provide more interpretation of the general versions of Le Cam's and Assouad's bounds (Theorems 1 & 3)? Even some simple intuition such as, when would we expect the lower bound to be non-negative, etc. would be helpful.

2. Can you more context for the parameter $\kappa$ and how it interpolates between the non-private and $\epsilon$-LDP settings?

3. Can you provide some intuition as to why your techniques for univariate mean estimation can't be easily extended to higher dimensions?

**Limitations:**

The authors have a nice discussion of the paper’s limitations (as well as possible directions for future work) at the end of their conclusion section. The main limitations include some restrictive technical conditions, as well as the fact that their mechanisms only achieve the minimax rate in limited settings.

---

> ### Author Rebuttal · Authors · 2023-08-09
>
> We appreciate the reviewer's thoughtful comments and insightful feedback.
>
> **Q1:**  Can the authors provide more interpretation of the general versions of Le Cam's and Assouad's bounds (Theorems 1 & 3)? Even some simple intuition such as, when would we expect the lower bound to be non-negative, etc. would be helpful.
>
> **A1:** The general form of Le Cam's and Assouad's inequalities could indeed become trivial, i.e., the risk bound can be negative. For the Le Cam's method as an example, a negative bound can be prevented by carefully choosing the two distributions $P_1$ and $P_2$ with "similar'' densities (i.e., smaller total variation) for a fixed value of $\eta$. A proper choice of these distributions will adequately represent the innate challenge regarding the given estimation problem quantified by the minimax risk. The choice of $\eta$ in relation to $n$ is made deliberately to ensure that the lower bound remains positive and it consequently influences the rate of the lower bound.
>
> The lower bound itself can be interpreted as the following.  For a given parameter difference $\eta$, as the dissimilarity between the densities (TV) decreases, the problem becomes difficult so the lower bound increases, and vice versa.
>
> **Q2:** Can you add more context for the parameter $\kappa$ and how it interpolates between the non-private and $\epsilon$-LDP settings?
>
> **A2:**  The role of the parameter $\kappa$ can be understood in the context of the contraction coefficient $c_{f,\kappa}=(1-\kappa)^{1-\kappa}\kappa^\kappa\int (\kappa+1)t^{\kappa-1}\delta_f(t)dt$, which involves $\delta_f(t)$
>     and $\kappa$. (Recall: $f$-FLDP is equivalent to $(\epsilon,\delta_f(e^\epsilon))$-LDP.) Our minimax bounds are non-trivial only when $c_{f,\kappa}$ is finite, and this necessitates $\delta_f(t)$ to decrease faster than $t^{-\kappa}$. Consequently, a larger value of $\kappa$ (close to 1) requires a smaller value of $\delta_f(t)$, aligning with a more private setting. On the contrary, in a less private setting, $\delta_f(t)$ grows, compelling $\kappa$ to decrease in order to uphold the condition $c_{f,\kappa}<\infty$.
>
> **Q3:** Can you provide some intuition as to why your techniques for univariate mean estimation can't be easily extended to higher dimensions?
>
> **A3:** The challenge in extending to the high-dimensional mean estimation problem is that we rely on the *probability* that the likelihood ratio is bounded, while in the case of $\epsilon$-LDP, it regards the pointwise-bounded nature of the likelihood ratio between outputs of mechanisms. This discrepancy introduces a challenging technical aspect, as the meticulous selection of distributions highlighted in A1 must be carried out for each component. Consequently, this entails managing multiple distributions in high-dimensional spaces. We suggest this problem as future research.

---

> > ### Comment · Reviewer_fz6L · 2023-08-14
> >
> > Thanks to the authors for their thorough response, I believe it has improved my understanding on several points. I maintain my score and hope to see this paper accepted.

---

### Official Review · Reviewer_qJiE · 2023-07-08

**Soundness:** 4 excellent
**Presentation:** 3 good
**Contribution:** 2 fair
**Rating:** 6
**Confidence:** 4

**Summary:**

The paper proposes a local version of functional differential privacy (FDP), and finds the minimax rate of mean estimation and non-parametric density estimation under local FDP. This paper amounts to an extension of the main results of Duchi et al. (2018) from epsilon-LDP to local FDP.

**Strengths:**

Originality: the paper is the first to consider extending functional DP to the local setting, to the best of my knowledge.

Quality and clarity: the technical claims are carefully proved and concisely explained.

Significance: because FDP offers many advantages over (epsilon, delta)-DP in the central setting, it is useful to understand whether some advantage also exists in the local setting. The comparison between mu-GDP and epsilon-LDP in this paper provides some (perhaps negative?) insight in this regard.

**Weaknesses:**

* Key concepts such as epsilon-LDP and local FDP are never formally defined in the main paper. As the definition of LDP is not entirely a trivial extension of central DP to begin with (for example, see Duchi et al. (2018)'s treatment of this concept), it is a risky choice to expect readers to extrapolate the (central) FDP definition to local FDP, as understanding the definition of local FDP is essential for reading the rest of the paper.

* High-level contribution of this paper. While I appreciate the neat mathematical arguments and results on local FDP, this paper leaves me wondering why local FDP is a worthwhile notion of privacy to consider. The original FDP paper by Dong et al. provides some convincing arguments for FDP, but this paper seems to be much more interested in deriving minimax rates as a mathematical exercise, without critically thinking about the underlying notion of privacy. I wish the paper had devoted some discussion to this question either before or after its mathematical investigations. From a mathematical point of view, the generalizations of epsilon-LDP Le Cam's and Assouad's inequalities by Duchi et al. can be interesting in their own right, but neither do these generalizations suggest any qualitative difference between epsilon-LDP and local FDP besides different contraction constants.

**Questions:**

* The role of lower bounds varying continuously with $\kappa$. If ultimately the matching upper and lower bounds only needed the $\kappa = 1$ case, what is the purpose of considering this continuous family of lower bounds? For those trade-off functions satisfying Lemma 1, does $\kappa = 1$ always imply the best minimax rate? Do you know of any trade-off function $f$, possibly violating Lemma 1, such that choosing a $\kappa$ strictly between 0 and 1 is necessary?

* What technical innovations are needed to extend the epsilon-LDP versions of Le Cam's inequality and Assouad's inequality to local FDP?

* From a DP practitioner's point of view, why might one want to consider local FDP? As the paper reveals, there is hardly evidence that local FDP offers better statistical utility, and the cognitive burden of describing/understanding local FDP is certainly higher than the epsilon-LDP case. Additionally, for non-Gaussian tradeoff functions, it may not be easy to come up with an appropriate privatization method.

**Limitations:**

Section 5 has comprehensively discussed the limitations from a technical point of view. Discussions on more foundational questions, such as the usefulness of extending FDP to the local setting, or even the technical innovations required, if any, for extending epsilon-LDP theory to local FDP, would be much appreciated.

---

> ### Author Rebuttal · Authors · 2023-08-09
>
> We appreciate the reviewer's thoughtful comments and insightful feedback.
>
> **Answer to Q1 on $\kappa$:** (i)  We examined a continuous range of lower bounds relative to $\kappa$ with the purpose of investigating the *gradual shift* in optimal utility associated with privacy constraints. It was motivated by the connection between the trade-off function and the contraction coefficient $c_{f,\kappa}$, as well as the exponent of the minimax rate. While our findings do not yield an explicit result for $\kappa<1$, we note that there has not been any research on private nonparametric density estimation in this particular $\kappa$ regime. Therefore, our work can provide indirect insights into private estimation with privacy constraints $\kappa<1$.
>
> (ii) It is true that $\kappa=1$ is optimal with respect to minimax rates for trade-off functions satisfying Lemma 1. Another way to appreciate this is the following: For a trade-off function $f$, there exists a trade-off function $f_{\epsilon}$ such that $f$-FDP is equivalent to $\epsilon$-DP for some $\epsilon>0$ except for the case of $f(x)=1-x$. This allows us to leverage optimal mechanisms designed for $\epsilon$-LDP, which achieve the minimax rates of $O\left(n^{-\frac{2k-2}{2k}}\right)$ for univariate mean estimation and $O\left(n^{-\frac{2\beta}{2\beta+2}}\right)$ for nonparametric density estimation. When $\kappa=1$, the corresponding lower bound retains the same rate, meaning that the theoretical lower bounds align with the optimal minimax rates.
>
> (iii) There are trade-off functions that contradict Lemma 1 and need $\kappa< 1$. For example, when $f(x)=1-x^{\frac{\kappa}{1+\kappa}}$ we have
> $\delta_f(y)=\sup_{x\in [0,1]}1-yx-f(x)=\sup_{x\in[0,1]}x^{\frac{\kappa}{1+\kappa}}-yx=\frac{\kappa^{\kappa}}{(1+\kappa)^{1+\kappa}}y^{-\kappa}$
> for large $y>y_0$ for some $y_0>0$. Also $1\geq f(y)\geq 1-y\cdot 0-f(0)\geq 0$. Thus, $\int t^{\kappa-1} \delta_f(t)dt$ diverges. Therefore, for every $\kappa_0\in (0,1)$, there exists a trade-off function requiring $\kappa$ to be less than $\kappa_0$.
>
> **Answer to Q2 on technical innovation:** Both Le Cam's and Assouad's inequalities establish minimax lower bounds by considering distributions that are similar but differ in target parameters. When these similar distributions possess substantially distinct parameters, the estimation task becomes challenging and the minimax risk tends to increase. Introducing a privacy constraint to the estimation problem yields that the distributions of observables—outputs of the privacy mechanism—become less distinguishable from one another compared to the original distributions, while the true value of the target parameter stays the same.
>
> To extend the Le Cam and Assouad methods from non-private settings to private estimation, one needs to establish a connection between the difference in distributions of mechanism outputs, $M(P_1)$ and $M(P_2)$, and the divergence between the original input distributions $P_1$ and $P_2$. Prior approaches achieved this via a uniform bound on the output distribution difference (expressed as the likelihood ratio) over the sample space, under the $\epsilon$-LDP assumption.
>
> In our work, we found that most local DP mechanisms tend to exhibit properties akin to $\epsilon$-LDP for certain values of $\epsilon$, even if they do not satisfy strict $\epsilon$-LDP.  In order to extend the $\epsilon$-LDP versions of Le Cam's and Assouad's inequalities to the local version of $f$-DP, we determined bounds for the probability that an $f$-DP mechanism behaves like $\epsilon$-DP.
> In other words, we derived an inequality that governs the likelihood ratio of outputs of mechanisms $\mathbb{P}\left(\frac{f_{M(P_1)}(Z)}{f_{M(P_2)}(Z)}>e^\epsilon|Z\sim M(P_2)\right)$ for distributions $P_1,P_2\in\mathcal{P}(\mathcal{X})$. Considering such probability was critical in order to treat local FDP as if they were $\epsilon$-LDP and this can be considered as a key technical innovation of our paper.
>
> **Answer to Q3 on local FDP:** We think that there might be a possibility for achieving a better minimax rate through local FDP compared to $\epsilon$-LDP. Specifically, in our response to the first question from Reviewer 6GNC, we theoretically demonstrate that our FLDP algorithm has an improved constant compared to the LDP algorithm Duchi et al. (2018) in nonparametric density estimation (even though the orders of the minimax rates are the same), which implies a practical advantage of our method. You can find more details regarding this discussion in that response.
>
> In addition, implementing privacy mechanisms under $\epsilon$-LDP presents a challenge in managing the privacy budget, particularly when composing multiple mechanisms. Effective composition rules are essential to strike the right balance between privacy protection and estimation accuracy. Inefficient composition rules tend to overestimate privacy leakage, leading to excessive perturbation of estimations. Even when minimax rates remain identical across different privacy schemes, an inefficient composition rule can practically undermine the overall performance. The shortcomings of the composition rule for $(\epsilon,\delta)$-DP are widely acknowledged, and other relaxations of DP have also faced criticism for their inefficient composition rules. In that regard, we believe that FDP possesses an effective composition rule.
>
> Lastly, we note that Awan et al. (2023) have introduced an additive mechanism that achieves $f$-FDP across a wide array of trade-off functions $f$, and Awan et al. (2022) have developed multivariate $f$-FDP mechanisms as well as log-concave $f$-FDP mechanisms tailored to specific $f$ functions. While we cannot definitively ascertain their suitability as effective privatization methods without further investigation, these approaches offer potential avenues for applying non-Gaussian trade-off functions to achieve privacy objectives.

---

> > ### Comment · Reviewer_qJiE · 2023-08-20
> >
> > Thank you for the detailed answers. They have certainly improved my understanding of the technical results.
> >
> > I will maintain my score. After reading the reviews and rebuttals, I believe that a more thorough comparison with local $\varepsilon$-DP can help improving both clarity and impact of the paper.

---

### Official Review · Reviewer_6GNC · 2023-07-26

**Soundness:** 4 excellent
**Presentation:** 4 excellent
**Contribution:** 3 good
**Rating:** 7
**Confidence:** 3

**Summary:**

This paper investigates the minimax risk achieved under functional local differential privacy (FLDP) constraints, and particularly under Gaussian local differential privacy (GLDP). The authors first introduce lower bounds for univariate mean estimation under FLDP using Le Cam’s method. Under certain assumptions on the threshold function (satisfied by GLDP) this rate matches the one with $\epsilon$-LDP introduced by Duchi et al (2018). They then use this result to derive upper and lower bounds for GLDP and a mechanism achieving this rate.

Similarly, they use Assoud’s method to derive lower and upper bounds for non-parametric density estimation, and the corresponding mechanism achieving optimality.

The authors empirically show their private mean estimation algorithm outperforms the one introduced in Duchi et al. (2018).


**Strengths:**

- The paper is very well written and clear. It introduces simple mechanisms to achieve optimal minimax rates for two important problems, namely (1) univariate mean estimation and (2) non-parametric density estimation.


- Further, their theoretical results provide an understanding of the tradeoff between utility and privacy for mean estimation. By analyzing the minimax risk under the lens of FLDP they introduce a continuous measure between privacy and utility, parametrized by a constant kappa, such that kappa=0 corresponds to non-privacy and kappa=1 corresponds to pure local DP.


**Weaknesses:**

1. Constants matter in differential privacy. In the current state of the paper it is clear that asymptotically the rates are the same, however it is hard to get an intuition of the constants. The plots however provide evidence that the suggested approach does provide better results.

2. After Duchi et al. 2018’s paper there have been other papers introducing local DP algorithms, however the paper only compares theoretically and empirically to Duchi et al. 2018. Due to constants these other algorithms could perform better under certain regimes.
- https://ieeexplore.ieee.org/document/8006630
- https://proceedings.mlr.press/v162/asi22b/asi22b.pdf

3. Algorithm complexity is only qualified as “less complex” or “more straightforward”.


**Questions:**

- Could the authors discuss the three main points above?


**Limitations:**

- The authors clearly describe some of the limitations of their work in the conclusion. First, the optimality is only achieved for kappa=1 that corresponds to pure local DP. Second, results only hold for a certain class of threshold functions. And finally, mean estimation is only analyzed in one dimension. The high dimensional case remains an open problem under FLDP.

- Besides these limitations, the condition on threshold functions does not seem easy to verify. Further, it remains unclear if in practice the mechanisms provided will actually have better performance.

---

> ### Author Rebuttal · Authors · 2023-08-09
>
> We appreciate the reviewer's thoughtful comments and insightful feedback.
>
> **Q1:** Constants matter in differential privacy. In the current state of the paper it is clear that asymptotically the rates are the same, however it is hard to get an intuition of the constants. The plots however provide evidence that the suggested approach does provide better results.
>
> **A1:** We agree that constants can have substantial meaning in differential privacy, especially in practice. We have attempted to derive the optimal constants for the results in Duchi et al. (2018) to compare with ours. Their result implies that
> \begin{equation*}
> \mathcal{R}\leq(\beta+1)\left(\frac{\beta}{\sqrt{2\pi e}}n\left(\frac{e^{\epsilon}+1}{e^{\epsilon}-1}\right)^{-2}\right)^{-\frac{2\beta}{2\beta+2}}r^{\frac{2}{\beta+1}},
> \end{equation*}
> while our risk bound from a similar calculation becomes:
> \begin{equation*}
> \mathcal{R}\leq(\beta+1)\left(0.5\beta n\mu^2\right)^{-\frac{2\beta}{2\beta+2}}r^{\frac{2}{\beta+1}}+O\left(n^{-\frac{2\beta+1}{2\beta+2}}\right).
> \end{equation*}
> Comparing the coefficients of $n^{-\frac{2\beta}{2\beta+2}}$, we have
> \begin{equation*}
>  \frac{c_{ours}}{c_{Duchi}}=\left(\frac{0.5\mu^2}{\frac{1}{\sqrt{2\pi e}}\left(\frac{e^{\epsilon}+1}{e^{\epsilon}-1}\right)^{-2}}\right)^{-\frac{2\beta}{2\beta+2}}=\left(\sqrt{\frac{\pi e}{2}}\mu^2\left(\frac{e^{\epsilon}+1}{e^{\epsilon}-1}\right)^2\right)^{-\frac{2\beta}{2\beta+2}}.
> \end{equation*}
> If $\mu=1$, as in the experiment, $c_{ours}<c_{Duchi}$ holds for every $\epsilon>0$, suggesting that our algorithm can potentially achieve a smaller risk, which is also empirically demonstrated in our paper. However, we note that these constants could only reflect the upper bounds of potential risks, rather than the risks themselves.
>
> **Q2:** After Duchi et al. (2018)’s paper, there have been other papers introducing local DP algorithms, however, the paper only compares theoretically and empirically to Duchi et al. (2018). Due to constants, these other algorithms could perform better under certain regimes.
>
> Ye and Barg (2017) (https://ieeexplore.ieee.org/document/8006630)
>
> Asi et al. (2022) (https://proceedings.mlr.press/v162/asi22b/asi22b.pdf)
>
> **A2:**  We agree that other algorithms could perform better under some regimes. However, we want to point out that our settings for the considered estimation problems are different from those works. First, the distribution family that are considered for nonparametric density estimation in both ours and Duchi et al. (2018) is assumed to have a density with smoothness parameter $\beta>1/2$. However, Ye and Barg (2017) and most other related works deal with either discrete distributions or a discretized version of a continuous density.
>
> Second, for the univariate mean estimation, we assume that the random variable is unbounded, but has bounded moments. The majority of related studies including Asi et al. (2022) focus on high-dimensional, bounded variables. This disparity makes the comparison unaccountable. Specifically, when privately estimating the mean of an unbounded random variable, a cut-off procedure is crucial to ensure privacy, and determining the cut-off point plays a significant role in achieving the optimal risk. In contrast, private mechanisms for bounded high-dimensional mean estimation do not need a cut-off, as their data is inherently bounded. The assumed boundedness yields the minimax rates that are comparable (with respect to $n$) to non-private estimation, which is not possible in our setting.
>
> **Q3:** Algorithm complexity is only qualified as “less complex” or “more straightforward”.
>
> **A3:** The computational complexity of both Duchi et al. (2018) and our algorithms is $O(n^{(\beta+2)/(\beta+1)})$. However, our algorithm is considered canonical and offers a more straightforward implementation.

---

> > ### Comment · Reviewer_6GNC · 2023-08-18
> > **Thanks!**
> >
> > Thanks for all the clarifications, I think adding a clarification on (1) could reinforce the merits of this paper. I maintain my score and I look forward to discussing it with other reviewers.

---

### Decision · Program_Chairs · 2023-09-21

**Decision:**

Accept (poster)

**Comment:**

All reviewers agreed that the paper should be accepted. The reviewers' questions during the discussion period led the authors to refine the constants and rates in their analysis, and they are encouraged to incorporate this discussion into the final version of their paper.